# Simultaneous Hand Gesture Classification and Finger Angle Estimation via a Novel Dual-Output Deep Learning Model

**DOI:** 10.3390/s20102972

**Published:** 2020-05-24

**Authors:** Qinghua Gao, Shuo Jiang, Peter B. Shull

**Affiliations:** The State Key Laboratory of Mechanical System and Vibration, School of Mechanical Engineering, Shanghai Jiao Tong University, Shanghai 200240, China; gaoqinghua@sjtu.edu.cn (Q.G.); jiangshuo@sjtu.edu.cn (S.J.)

**Keywords:** modified barometric sensor, gesture recognition, convolutional neural network, dual-output model

## Abstract

Hand gesture classification and finger angle estimation are both critical for intuitive human–computer interaction. However, most approaches study them in isolation. We thus propose a dual-output deep learning model to enable simultaneous hand gesture classification and finger angle estimation. Data augmentation and deep learning were used to detect spatial-temporal features via a wristband with ten modified barometric sensors. Ten subjects performed experimental testing by flexing/extending each finger at the metacarpophalangeal joint while the proposed model was used to classify each hand gesture and estimate continuous finger angles simultaneously. A data glove was worn to record ground-truth finger angles. Overall hand gesture classification accuracy was 97.5% and finger angle estimation $R^{2}$ was 0.922, both of which were significantly higher than shallow existing learning approaches used in isolation. The proposed method could be used in applications related to the human–computer interaction and in control environments with both discrete and continuous variables.

## 1. Introduction

With the rapid development of computer technology and the widespread use of sensors, there is an emerging demand for more intuitive human–computer interaction (HCI) [1]. HCI approaches have been investigated for various applications, including touch interaction, voice control, and emotion perception. Gesture recognition is a promising HCI area, including a wide range of potential applications: sign language recognition [2], navigation and manipulation in virtual environments [3], automotive manufacturing [4], and control for home service robots [5]. The hands are a logical choice for gesture recognition because of their dexterity, and intricate movements involving multiple degrees of freedom [6,7,8].

Hand gestures can be classified as static or dynamic. Static hand gesture recognition considers the shape characteristics of the hand at a point in time, and dynamic hand gestures focus on a series of hand movements over a time interval. Dynamic hand gestures can involve both hand gesture classification and finger angle estimation. Hand gesture classification can decode human intentions as discrete commands for interacting with and controlling physical hardware. Finger angle estimation can be utilized to achieve the continuous operation of physical devices. For example, varying finger angles could be mapped to operate dynamic realistic tasks, such as game joysticks in virtual reality.

Various machine learning-based algorithms have been proposed for hand gesture classification. Jiang et al. [9] recognized four surface gestures and eight air gestures based on a fusion of surface electromyography and inertial measurement unit signals, utilizing linear discriminant analysis (LDA). Zhang et al. [10] presented Tomo, which employed electrical impedance tomography to evaluate two gesture sets with a support vector machine (SVM). Jiang et al. [11] classified singer-finger gestures and American Sign Language 0−9 number gestures with LDA, k Nearest Neighbor (KNN), and random forests (RF). Deep learning has also been studied in hand gesture classification. Wang et al. [12] proposed a combination of convolutional and recurrent neural networks (RNN) with Google’s Soli sensor to recognize 11 dynamic gestures. Kim et al. [13] used Doppler radar to recognize hand gestures with a convolutional neural network (CNN).

Machine learning approaches have also been proposed for finger angle estimation. Kawaguchi et al. [14] detected finger movement from wrist deformation and used multiple regression models to estimate finger joint angles. Zhou et al. [15] estimated finger motion through forearm muscle deformation measured by A-mode ultrasound-sensing approaches with a linear ridge model and a least-squares SVM model. Deep learning has also been investigated in the continuous angle estimation of fingers. Smith et al. [16] asynchronously decoded metacarpophalangeal (MCP) joint angles of five fingers using a backpropagation neural network classifier with a Levenberg–Marquardt training algorithm. Hioki et al. [17] estimated proximal interphalangeal finger joint angles by streaming feedback into a recurrent artificial neural network with time delay factors.

Some have also proposed simultaneous hand gesture classification and finger angle estimation. Pan et al. [18] proposed a switching regime, including one LDA classifier to identify wrist movements and two state space models to continuously decode the finger angle from electromyogram (EMG) signals. Antfolk et al. [19] classified finger movements and produced finger joint angle outputs using local approximation and lazy learning, consequently controlling the prosthetic hand. 

Some studies simultaneously output the results of classification and regression for other areas. Abraham et al. [20] presented a joint regression and classification framework to predict the future values of a zero-inflated time series. A linear regression model was used to predict the value of the time series, and the regression results were input to the SVM classifier to determine whether the predicted value should be adjusted to zero. Wang et al. [21] proposed a hybrid CNN model integrating a classification network and a regression network for object pose estimation from a single image, but their algorithms were based on static image data and thus not suitable for low-dimensional and dynamic time-series signals in our application.

However, current approaches suffer from several limitations. First, previous machine learning models produced limited accuracy because features were typically extracted independently from time-series signals in an ad-hoc manner [22], and the correlations among different signals were often overlooked. Although deep learning avoids the tedious process of manually extracting features, it relies on proper data structures. If the dimension of data is too small, the model may overfit and lead to low accuracy and poor performance. Second, shallow learning approaches cannot output classification values and regression values simultaneously. Although some articles investigated the output of hand gesture categories and finger angle estimation at the same time, they still relied on two superimposed algorithm models: one for classification and the other for regression. But this pattern has inherent drawbacks: (1) since the model parameters are derived from the data features, the attributes of multiple models are statistically relevant, and two separate models may create system redundancies. (2) When data changes, the classification and regression models require retraining, and the parameters of the two models cannot be adjusted simultaneously. (3) In the practical prediction applications, compared to the parallel structure that simultaneously derives the category and the regression values, the ordinal structure of first predicting the category and then estimating the angle values increases the complexity and inconvenience of the system.

Therefore, an improved compact model that can fully explore the spatial-temporal features should be built to enable two simultaneous outputs. To the best of our knowledge, there are currently no approaches for simultaneous outputs of hand gesture classification and finger angle estimation based on deep learning.

The purpose of this paper is to address the previous two limitations by introducing a novel framework to simultaneously detect hand gestures during finger movement with high accuracy, including identifying which finger is flexing and determining the joint angle of each finger. To improve the accuracy of the model, we present a new spatial-temporal method to explore the hidden relationships between different sensors and useful information contained in historical data. For simultaneous hand gesture classification and finger angle estimation, we propose a dual-output model structure. We hypothesized that the framework would enable classification of the hand gestures and estimation of finger angles simultaneously via a single deep learning model with dual outputs, and this framework would have better performance than single-output, shallow learning models.

## 2. Materials and Methods

### 2.1. Wristband Prototype and Experiment Protocol

#### 2.1.1. Wristband prototype

While current wearable approaches are limited because of low sensitivity or poor linearity, our prototype takes advantage of the fact that hand motion changes the shape of the wrist due to the expansions and contractions of the muscles, which brings position and thickness transformations of tendons and muscles, producing deformations on the surface contour of the wrist. We thus used a wristband with ten embedded modified barometric sensors with high sensitivity and excellent linearity to measure the activity of the tendons and muscles that cause finger movement [23]. The deformations apply pressure to the surface of modified barometric sensors. When a subject with the wristband performs various gestures, a corresponding contact pressure distribution can be obtained from the wristband (Figure 1).

#### 2.1.2. Gesture Set

When handling actual tasks by hand, multiple fingers often work in concert. The Taiwanese Sign Language includes 50 fundamental gestures [24], and many of them are the linkage of multiple fingers. Most studies about interlocking multiple fingers used static positions. However, we studied the dynamic process of finger movement. The dynamic recognition process of multiple fingers is complicated because the flexion angles of different fingers are different, and it involves a lot of research on hand anatomy. Multi-finger movements include thumb opposition, multi-finger flexion/extension, multi-finger adduction/abduction, etc. These multi-finger movements can be decomposed into multiple single-finger movements. Analyzing single-finger movements is the basis of multi-finger movements. At present, most studies on finger movement are based on single finger movement [15,25]. Therefore, we chose 5 single fingers as the gesture set to identify the moving fingers and estimate the finger joint angle.

#### 2.1.3. Experiment Protocol

Ten subjects (8 male, 2 female, age: 22–26 year, height: 166–184 cm, weight: 57–76 kg, wrist circumference: 15.3–16.9 cm) participated in the experiment. All subjects gave their informed consent in accordance with the Declaration of Helsinki before they participated in the study. During each trial, subjects flexed/extended one finger at the MCP joint repeatedly at their self-selected frequency. The minimum and maximum angles of thumb were required to be 0° and 55°, and of other 4 fingers were required to be 0° and 90°. Some subjects could not bend their fingers to 90° [26]. Therefore, the actual maximum and minimum angles were decided by the subjects themselves. Fingers were flexed in the order of thumb, index finger, middle finger, ring finger, and pinky finger, and the data were encoded into 5 categories and recorded. We used a data glove (5DT Data Glove 14 Ultra, South Africa, Gauteng, Pretoria), which is widely used and regarded as the gold standard for measurement in the estimation of finger joint angles, to record the real finger angle relative to the ground at the same time [14]. Each finger was flexed/extended for 25 s, there were 5 fingers per trial (Figure 2), and the sampling frequency was 40 Hz, so 25×5×40=1000 samples were collected per trial. Each subject performed 10 trials [27], so 10×10=100 trials were performed. Thus, there was a total of 1000×100=100,000 samples. 

For each trial, we had 10 columns of modified barometric sensing data, corresponding finger categories and finger angle data. The data dimension was (1000, 12). Given these data, we aimed to learn a mapping f from the pressure sensing data x∈X to dual outputs $\left[y_{1},\;y_{2}\right]\in Y$, where $y_{1}$ represents the label of the finger, totaling 5 classes, and $y_{2}$ represents the corresponding MCP angle measured by the data glove.

### 2.2. Data Augmentation

Sequential data were usually represented as a multivariate time series, and so the 10 columns of modified barometric sensing data were encoded as 10 features. However, there are restrictions on this data form. According to our sensing scheme and physiological structure of users, spatial locations between different sensors are related, but multivariate time-series form cannot distinguish a relationship between sensors that are far away. Suppose there are N-channel signals arranged in columns. For the column k signal, it is only adjacent to the column k−1 signal and column k+1 signal. When referring to the spatial information of the column k signal, only the correlation between column k signal and the column k−1 signal, and the correlation between the column k signal and column k+1 signal are considered, while the spatial positional relationships with signals of other columns are ignored. Therefore, we propose a data augmentation method to explore the hidden spatial–temporal relationships between different sensors.

We propose to convert time-series signals collected from sensors to new movement images that contain hidden relationships between any pair of signals (Figure 1). Features would be extracted from the newly generated movement image, providing additional potential spatial information between the signals.

The algorithm steps are as follows:N-channel signals are arranged into a 2D array L×N, where L is the time length of signal sequences, which is the original signal array.The 2D array from Step 1 is permutated into L×M, where M is the dimension of a newly generated array so that each signal sequence is adjacent to every other sequence [28]. For example, 3-channel signals [1, 2, 3], wherein the column 3 signal and column 1 signal are not adjacent, can be rearranged into 2D array signals [1, 2, 3, 1]. Then, each column of signals can be adjacent to other columns of signals, which takes into account the spatial information between each column of signals, and M=4 in this example.The new signal array from Step 2 is converted into a movement image by windowing, and the image size is Ls×M, where Ls is the length of the window.The window slides along the time axis to generate different movement images that are utilized as new inputs of the model.

Our signal data have 10 signal sequences [1, 2, 3, 4, 5, 6, 7, 8, 9, 10]. After Step 1 and Step 2, a movement image is generated with the size of 42×Ls, where Ls is the time length of signal sequences. For each trial, Ls=1000. The 42-dimensional vectors are [1, 2, 3, 4, 5, 6, 7, 8, 9, 10, 1, 3, 5, 7, 9, 1, 4, 6, 8, 10, 2, 4, 7, 10, 3, 6, 9, 2, 5, 8, 1, 5, 9, 3, 7, 1, 6, 10, 4, 8, 2, 6], and each column from 1 to 10 is adjacent to the other nine columns. A time window is utilized to crop the signals based on Step 3. The window length is 42, and the size of our movement image is finalized as 42×42. The window slides 1-time step along the time axis to generate a new movement image, and after Step 4, Ls−42 images are ultimately obtained. These images are new samples of the model, and the dimension of the samples is (Ls−42, 42, 42).

For each newly generated movement image, there are 42 finger labels and angles corresponding to window length. We select the finger category with the highest frequency as the label for this movement image, and the finger angle of the last time step as angle regression value for this image. For speeding up the convergence, the sensor features and finger angle data are standardized with a mean of 0 and a standard deviation of 1.

### 2.3. Model Architecture

We proposed a deep learning model with dual outputs to enable the classification of finger gestures and the estimation of finger angles simultaneously (Figure 3). We built a dual-output model based on CNN. We used 10 sensor units to measure the pressure changes of 10 tendons related to the finger movement. The pressure can be converted into a force myography (FMG). CNN can be used to identify specific FMG patterns. CNN is a deep learning model that extracts local features of data through convolution operations and weight sharing and is widespread in computer vision and pattern recognition. 

#### 2.3.1. Convolutional Layer

The convolutional layer includes multiple convolutional filters that extract the signal features by the convolution process. The sparse connection and weight sharing of the convolution kernels allow the convolutional layer to learn features with a small amount of calculations.

An activation function is used after the convolutional filter to solve non-linear problems. Deep learning currently employs a Rectified Linear Unit (ReLU) function as activation function, f(x)=max(0, x). This function defines the non-linear output after the linear transformation of the neuron.

The pooling layer is used for reducing data dimensions and preventing network overfitting after the convolutional layer. Several convolutional layers and pooling layers are often stacked to extract gradually more abstract features.

#### 2.3.2. Dual Outputs

A flatten layer is utilized to transform a multi-dimensional array to a one-dimensional vector, which then flows into a dense layer to reduce the dimension of neurons. Finally, the model takes two parallel dense layers as outputs: one with K neurons to export the category attribute, and the other with 1 neuron to export the regression attribute, where K is the number of hand gesture categories (Figure 3).

The dense layer used for classification requires a K-way softmax activation function to obtain the probability distribution over different categories:(1)$$y_{i}=\frac{e^{z_{i}}}{\sum_{j=1}^{K}e^{z_{j}}},\;i=1,2,3,\ldots ,K$$
where z is the output vector of the dense layer, $y_{i}$ is the probability that the sample belongs to the category i. The category with the highest probability is the predicted label of the sample.

#### 2.3.3. Loss Weighting and Training Algorithm

Angle estimation is a scalar regression task, while hand gesture recognition is a multi-category task. The training model needs to assign different loss functions to the two outputs. The mean-squared error can be used as the loss of regression output:(2a)$$loss_{r}=MSE=\frac{1}{n}\sum_{i=0}^{n-1}{\left(y_{i}-\hat{y_{i}}\right)}^{2}$$
where $loss_{r}$ is the regression loss, $y_{i}$ is the true value of sample i, $\hat{y_{i}}$ is the predicted value of sample i, and n is the number of samples.

Categorical cross entropy can be used as the loss of classification output:(2b)$$loss_{c}=J=-\frac{1}{n}\sum_{i=0}^{n-1}\sum_{j=0}^{K-1}y_{i,j}\log\left(p_{i,j}\right)$$
where $loss_{c}$ is the classification loss, $y_{i,j}\in \left(0,\;1\right)$ indicates whether the sample i is predicted to be the label j, $p_{i,j}$ is the probability that the sample i is predicted to be the label j, and K is the number of hand gesture categories.

Two losses need to be combined into a total loss. The scale of the two losses is different, which leads to the total loss being dominated by one of the tasks, and the loss of other task cannot affect the learning process of the network. A practical way to solve the problem is loss weighting, and thus, we weighted the losses to get a global loss. The goal of model optimization is to minimize total loss during training.
(2c)$$loss_{total}=w_{c}loss_{c}+w_{r}loss_{r}$$
where $loss_{total}$ is a total loss, $w_{c}$ is the weight of classification loss, and $w_{r}$ is the weight of regression loss. The ratio of $w_{r}$ to $w_{c}$ is the weight ratio, which is a hyperparameter.

The coefficients of convolution filters and dense layers are obtained by training a dataset. The dual-output model is multi-task learning based on parameter sharing. Multiple tasks share the representation at shallow layers, which can reduce network overfitting and improve the generalization effect.

Our dual-output CNN model was established according to the principle described above and was trained in Keras 2.2. The first convolutional layer filtered the 42 × 42 input movement image with 32 kernels of size 3 × 3, followed by a max-pooling layer of size 2 × 2. The second convolutional layer filtered the output of the first pooling layer with 32 kernels of size 3 × 3, followed by a 2 × 2 max-pooling layer. The flatten layer vectorized the output into a 128-dimension feature vector. The dense layer consisted of 32 units. The network ended with a 5-way dense layer as classification output and a 1-way dense layer as the regression output. We adopted: (1) a ReLU activation function in each hidden layer, (2) a Softmax function in the 5-way dense layer, (3) a 4:1 weight ratio between the finger angle regression loss and the finger gesture classification loss, and (4) an RMSprop function as the optimizer with a batch size of 32 and 10 epochs in model training.

### 2.4. Model Performance Evaluation

Leave-one-trial-out cross validation was utilized to assess the performance of the model. For each subject, the network was trained based on his training subset and evaluated based on his test subset. Each subject had 10 sets of experimental data, and we conducted 10 verifications for each subject. Each verification used 9 sets of data as the training set, the remaining set of data as the test set. Therefore, for the training set, the time length Ls=9×1000=9000, and the dimension of features was (9000−42, 42, 42). For the test set, the time length Ls=1000, and the dimension of the features was (1000−42, 42, 42). The test set selected for each verification was different from each other.

For each subject, the proportion of correctly recognized finger gestures in all test samples was used as the accuracy index of finger classification:(3)$$Accuracy=\frac{n_{correct}}{n_{total}}$$
where $n_{correct}$ is the number of samples classified correctly, and $n_{total}$ is the number of all test samples. The $R^{2}$ was used to access the accuracy of finger angle estimation:(4)$$R^{2}=1-\frac{\sum_{i}{\left(y_{i}-\hat{y_{i}}\right)}^{2}}{\sum_{i}{\left(y_{i}-\bar{y_{i}}\right)}^{2}}$$
where $y_{i}$ is the true value, $\hat{y_{i}}$ is the predicted value, and $\bar{y_{i}}$ is the mean value of test samples. The larger the $R^{2}$ score, the better the model fitting effect. The average accuracy over all subjects was calculated as the ultimate gesture classification accuracy and the average $R^{2}$ score over all subjects was calculated as the ultimate $R^{2}$ score.

### 2.5. Benchmarking against Other Machine Learning Models

For validating the performance of our proposed dual-output model, we established the following benchmarks: CNN without data augmentation, RNN, single-output classification models, and single-output regression models. SVM and RF can be utilized for both classification and regression, so they were chosen as the benchmarks of the single-output models. 

A paired *t*-test was used to assess whether there were significant differences between any benchmark and the proposed method. Since there were only 10 subjects, and the size of the samples was small, the data might not conform to the normal distribution. The nonparametric test does not require a population distribution, so a nonparametric paired *t*-test was ultimately selected. SPSS was used to perform the nonparametric paired *t*-test on the results of 10 subjects. When p<0.05, we considered it a significant difference between the two groups of samples.

#### 2.5.1. CNN without Data Augmentation

A dual-output CNN without data augmentation model was established to observe the impact of our proposed data augmentation method on the dual-output deep learning model. The original signal was composed of 10 columns of features, and we took a time window with a length of 10. The size of the movement image was finalized as 10×10, and the dimension of the samples was (Ls−10, 10, 10). Other selections, namely (1) the processing of finger labels and angles to keep up with each image, (2) the structures and parameters of the dual-output model, (3) the parameters of model training, were the same as the CNN with data augmentation model.

#### 2.5.2. RNN

RNN is a deep learning model with memory and is widely used to process time-series signals. Base on the gating mechanism, long short-term memory (LSTM) retains long-term sequence information and alleviates the gradient disappearance problem [29]. The gated recurrent unit (GRU) is a simplified variant of the LSTM [30]. The GRU network does not maintain additional state vectors and has less computation and faster training than the LSTM.

We built a dual-output model based on GRU to compare with the CNN dual-output model. The network structure and parameters were as follows: (1) the 10 columns of original signals were taken as the input, so the dimension of the input was 10, (2) the length of time step was 10, (3) the finger category with the highest frequency in 10-length time step was taken as the true label, and the last angle value in 10-length time step was taken as true angle value, (4) 1 GRU layer was used, (5) the dimension of the output was 32, (6) the dropout of input unit was 0.2, and the dropout of the recurrent unit was 0.2.

The GRU layer was followed by the dual-output structure, which was the same as the principle described in the dual-output CNN model. Other selections of the weight ratio, the optimizer, the batch size, and the epoch, were the same as the CNN model parameters.

#### 2.5.3. SVM

SVM is a widely adopted approach in the realm of hand gesture recognition. It uses hinge loss to calculate empirical risk and uses regularization to optimize structural risk. SVM can be used for classification with support vector classification (SVC) and regression with support vector regression (SVR). SVC is a classifier with sparsity and robustness. It can be nonlinearly classified by the kernel method. Its goal is to maximize the margin between support vectors and the hyperplane. Different from SVC, the goal of SVR is to minimize the total deviation of all sample points from the hyperplane.

We used the RBF kernel function and tuned the penalty coefficient (C) and the kernel parameter gamma (γ) to obtain higher accuracy and $R^{2}$ score. An SVC model with the parameters C=5 and γ=0.1 was established to classify finger gestures, and an SVR model with parameters C=10 and γ=0.1 was established to estimate the finger angle.

#### 2.5.4. RF

RF is a bagging-based ensemble learning method composed of multiple weak learners. RF can be used for classification with RF classifier and regression with RF regressor. RF classifier randomly generates k decision trees from the training set to form a random forest by the bootstrap resampling technique. The classification result is determined by decision tree voting scores. Different from classification, the RF regression result is the average of the outputs of decision trees.

We tuned the parameters of the number of decision trees and the maximum number of features selected when dividing leaf nodes. For the RF classifier, the number of decision trees was 100, and the maximum number of features was $\sqrt{n_{features}}$, where $n_{features}$ is the number of total features. For the RF regressor, the number of decision trees was 60, and the maximum number of features was ${log}_{2}n_{features}$.

### 2.6. The Impact of Sensor Numbers on the Dual-Output Model

We compared the results of different sensor numbers between 1 and 10. Tree-based feature selection was used to calculate the importance of features. The RF classifier and RF regressor were, respectively, applied as tree models to obtain the feature importance $\alpha_{c}$ on classification attributes and $\alpha_{r}$ on regression attributes. The mean value $\alpha_{i}$ of $\alpha_{c}$ and $\alpha_{r}$ for each subject was taken as the total importance of this subject, where i=1,2,…,10. We calculated the average of total importance for all subjects as the ultimate results of feature importance, which was sorted in reverse order as [8, 1, 3, 7, 5, 6, 2, 4, 9, 10]. We selected 1 to 10 features with the highest feature importance and used the proposed data augmentation method and dual-output model to obtain the results of different sensor numbers.

## 3. Results

The mean classification accuracy of hand gesture was 97.5% across the subjects. A confusion matrix was drawn from subject 10 (Figure 4), and there was no obvious confusion in the classification. The thumb had the highest classification accuracy at 100.0%, while the middle finger had the lowest classification accuracy at 97.5%. The predicted label typically confused the middle finger and ring finger. The mean $R^{2}$ score of the finger angle estimation over all subjects was 0.922. The fitting effect was better in the rising and falling phases of the curve, but there were deviations in the peaks and valleys (Figure 5).

The results (Figure 6) of four benchmarks were as follows: (1) CNN without data augmentation had a classification accuracy of 96.3% and angle $R^{2}$ score of 0.916. (2) RNN had a classification accuracy of 95.6% and an angle $R^{2}$ score of 0.881. (3) SVM had a classification accuracy of 96.5% and an angle $R^{2}$ score of 0.887. (4) RF had a classification accuracy of 95.8% and an angle $R^{2}$ score of 0.905.

The results (Figure 7) of different sensor numbers between 1 and 9 were as follows: (1) one sensor had a classification accuracy of 76.4% and an angle $R^{2}$ score of 0.528; (2) two sensors had a classification accuracy of 87.8% and an angle $R^{2}$ score of 0.729; (3) three sensors had a classification accuracy of 91.5% and an angle $R^{2}$ score of 0.820; (4) four sensors had a classification accuracy of 93.2% and an angle $R^{2}$ score of 0.855; (5) five sensors had a classification accuracy of 95.7% and an angle $R^{2}$ score of 0.889; (6) six sensors had a classification accuracy of 96.9% and an angle $R^{2}$ score of 0.909; (7) seven sensors had a classification accuracy of 97.3% and an angle $R^{2}$ score of 0.916; (8) eight sensors had a classification accuracy of 97.3% and an angle $R^{2}$ score of 0.918; (9) nine sensors had a classification accuracy of 97.4% and an angle $R^{2}$ score of 0.920.

## 4. Discussion

### 4.1. Comparison with Previous Results

This paper presents a novel framework to output hand gesture classification and finger angle estimation simultaneously via a deep learning model with dual outputs. Compared with previous methods, this scheme enables outputs of both classification and regression attributes simultaneously with higher accuracy. For hand gesture recognition with shallow learning, Al-Timemy et al. [31] achieved 98% accuracy for the classification of 15 classes of different finger movements with time-domain autoregression, orthogonal fuzzy neighborhood discriminant analysis, and LDA. For hand gesture recognition with deep learning, Geng et al. [32] used EMG images and a deep convolutional network for an 8-gesture within-subject recognition. The recognition accuracy reached 89.3% on a frame of the image and 99.0% using the majority voting algorithm over 40 frames. For finger angle estimation with shallow learning, Huang et al. [25] employed an ultrasound image to predict the MCP joint angles of all fingers in nonperiod movements and got an average correlation of 0.89 ± 0.07. For finger angle estimation with deep learning, Ngeo et al. [33] utilized a fast feedforward artificial neural network and a nonparametric Gaussian process (GP) regressor to capture finger angles during finger flexion and extension. This method yielded an $R^{2}$ of 0.78. For hand gesture classification and finger angle estimation simultaneously with two superimposed models, Yang et al. [34] reported a class-specific proportional control system and utilized LDA for gesture recognition and GP regression for muscle contraction force estimation via wearable ultrasound sensing. The accuracy of this model was 93.7%, and the $R^{2}$ was 0.927. Previous research [27] used the SVM, LDA, and KNN models for the recognition of three sets of static hand gesture, and an RF model for dynamic single finger angle estimation. Thus, the classification and regression attributes of gesture data were not considered jointly. In addition, previous research achieved a classification accuracy of 94.4% on a set of static single finger flexion gestures. However, we obtained a classification accuracy of 97.5% on a set of dynamic single finger flexion gestures. Both studies examined finger angle estimation based on a dynamic single finger flexion experiment, but we got a higher $R^{2}$ of 0.922 compared with 0.79 in previous research. Therefore, our model implemented more functions and exhibited higher accuracy.

### 4.2. Analysis of Confusion Matrix and Fitting Curve

The least frequent misclassification occurred in the thumb, while the most frequent misclassification occurred in the middle finger (Figure 4). This happens because the flexion and extension movements of the thumb are on the sagittal axis, while the flexion and extension movement of the other four fingers are on the coronal axis. Therefore, the flexion and extension of the thumb are rarely affected, and the classification is not easily confused with the other four fingers. When the middle finger was flexing and stretching, it was easily confused with the index finger and ring finger because the same ligament drives the index finger and ring finger at the same time.

Angle estimation did not fit as well at the peaks and troughs in some cases (Figure 5). The reason might be that the activities of the tendons and muscles that caused finger movement were very complicated when the fingers varied from a flexed state to a stretched state or vice versa, and the model could not statistically excavate the principles.

### 4.3. Effects of Hyperparameters

The effects of hyperparameters on the performance of the model were investigated.

#### 4.3.1. Window Length

The window length indicated the length of each movement image. The longer the window length, the more past information each image contained, which could improve the performance of the model. However, when the window length was too long, each image contained too much past information, which caused the model to be overburdened and make it overfit. Therefore, the appropriate window length should be selected. In the model training, we found that the model performance was best when the window length was between 40 and 50. Considering that the channel dimensions after data augmentation were 42, the window length was ultimately selected to be 42.

#### 4.3.2. Network Structure of the Model

We also assessed the impact of the network structure on the performance of the model and found that, with the increases in the model depth, the performance did not improve significantly. Therefore, a shallow network structure was suitable because a very large and deeper network structure would learn noise and other interference errors and could result in performance degradation or increased computational cost and network burden.

#### 4.3.3. Batch Size and Epoch

The batch size and the epoch followed a similar trend that, as they increased, the performance of the model decreased. So, a smaller batch size and epoch should be selected. The reason for the shallow network structure and small batch size and epoch might be that our data was less complicated, so a lightweight network was sufficient to extract features.

#### 4.3.4. Weight Ratio

The weight ratio was a hyperparameter introduced to make the scales of classification loss and regression loss close. In the model training, we adjusted the weight ratio from 20:1 to 1:20 and found fluctuations in performance. A possible reason for this is related to the decay rate of the loss function. In the model training, the decay rates of the two loss functions were inconsistent. Therefore, a fixed weight ratio might not be effective enough. In the weight ratios we tested, when the weight ratio was 4:1, the effect was best, so we used 4:1 as the final weight ratio.

### 4.4. Feature Extraction in Sequential Signals and Image Signals for CNN

#### 4.4.1. Feature Extraction in Sequential Signals for CNN

CNN can be well applied to sequential signals, such as inertial measurement units (IMU) signals and audio signals. Time can be viewed as a spatial dimension. The 1D convolution layer can extract and recognize local patterns in a sequence. Because the same transformation is performed on every patch, a pattern learned at a specific position can be recognized at a different position later, making 1D CNN translation-invariant for temporal translations.

#### 4.4.2. Feature Extraction in Converted Image Signals for CNN

When the time series data are converted into images, the convolutional kernel of 2D CNN slides on the image. The temporal information is extracted when the convolutional kernel slides from top to bottom on the image. The spatial information between different sensors is extracted when the convolutional kernel slides from left to right on the image.

When the convolutional kernel slides from top to bottom on the image, it is very similar to 1D CNN. When the convolutional kernel slides from left to right on the image, and the data of one sensor changes to the data of an adjacent sensor, a drastic change will occur, thus generating a high-frequency signal. Through the convolution operation between the convolution kernel and the high-frequency signal, the frequency information will be selected, so CNN can extract the position information between the sensors.

### 4.5. Comparison with Other Machine Learning Models

From the comparison of the dual-output models based on CNN with data augmentation and other models (Figure 6), we found that the model built by CNN with data augmentation performed best. This model had the highest accuracy, highest $R^{2}$ score, and the smallest deviations.

#### 4.5.1. Comparison between Dual-Output Models with and without Data Augmentation

From the comparison between the dual-output CNN models with and without data augmentation (Figure 6), for hand gesture classification, the CNN without data augmentation model had a classification accuracy of 96.3%, and the CNN with data augmentation model had a classification accuracy of 97.5%. A nonparametric paired *t*-test indicated significant differences between the CNN with data augmentation and the CNN without data augmentation. The data augmentation method not only made classification accuracy higher but it also made deviation smaller. For finger angle estimation, the CNN without data augmentation model had a $R^{2}$ score of 0.916, and CNN with data augmentation model had a $R^{2}$ score of 0.922. Although a nonparametric paired *t*-test indicated no significant difference between dual-output models with and without data augmentation, there was a smaller deviation in the CNN with data augmentation model.

The results show that the data augmentation method achieved better results. This is because the data augmentation method took into account the hidden relationships between different sensors and useful information contained in history data.

#### 4.5.2. Comparison between Dual-Output Models Based on CNN with Data Augmentation and RNN

From the comparison between dual-output models based on CNN with data augmentation and RNN (Figure 6), the RNN model had a classification accuracy of 95.6% for hand gesture classification and had an $R^{2}$ score of 0.881 for finger angle estimation. A nonparametric paired *t*-test indicated significant differences between the CNN with data augmentation model and RNN model both in hand gesture classification and finger angle estimation. CNN with the data augmentation model had a smaller deviation. Similarly, CNN without the data augmentation model was also better than the RNN model. Therefore, the models built by CNN performed better than RNN.

RNN is mainly utilized to describe the continuous output in time and to connect spatial states together. It is more suitable for predicting future data from past data, such as the prediction of the next moment of the video, the prediction of the context of the document, and the prediction of future temperature. CNN, however, is more suitable for describing the state of a certain space and feature extraction. Our research tends to perform feature extraction on data to identify patterns in gesture data, rather than predicting what gesture will be made or what angle the finger will bend at the next moment. 

RNN only considers the temporal information of the data and does not consider the spatial information. We have 10 sensors around the wrist, and there are positional relationships between the 10 sensors. CNN can extract the position information between the sensors. 

CNN is more appropriate for hand gesture classification and finger angle estimation by the above discussion. The results further show that although a dual-output model could be implemented based on deep learning, our proposed method had better results.

#### 4.5.3. Comparison between Dual-Output Models and Single-Output Models

From the comparison between the dual-output models based on deep learning and the single-output models based on other machine learning (Figure 6), we found that the performance of the dual-output models based on deep learning was not necessarily higher than the single-output model based on shallow learning, and sometimes even worse. For example, in terms of finger angle estimation, the result of CNN without data augmentation was higher than SVM and RF, but the result of RNN was much lower than RF. In terms of hand gesture classification, although the result of CNN was higher than RF, the result of RNN was much lower than SVM. However, in general, our proposed method was superior to shallow learning models in both hand gesture recognition and finger angle estimation.

The results of the above comparisons can verify our hypothesis. The newly proposed data augmentation method can discover hidden spatial position relationships in time-series signals and improve the accuracy of the results. Deep learning models can learn different attributes of the signal at the same time rather than in succession. When the model adjusts its internal features, other features associated with them can automatically adapt and adjust. This is more powerful than stacked shallow learning models.

### 4.6. Comparison of Results with Different Sensor Numbers

The classification accuracy and angle fitting degree improved with the increase in sensors (Figure 7). When using the optimal six or more sensors, the classification accuracy reached more than 96% and the $R^{2}$ value reached more than 0.9. When using the optimal three to five sensors, the classification accuracy reached more than 90% and the $R^{2}$ value reached more than 0.8. When using the optimal one or two sensors, the results dropped sharply, and the model performance was poor. Therefore, the performance of the model could be improved by increasing the number of sensors. However, when the optimal number of sensors exceeded six, the performance of the model improved more slowly, and the device became more complicated.

### 4.7. Calculation Time and Real-Time Application

In practical application, the time duration for each gesture was about 2 to 3s. Therefore, after the dual-output model was trained offline, we selected samples of 2 s in a test set (subject 10, trial 1) to calculate the prediction time. We got the prediction time 0.0444 s on the proposed dual-output model (Table 1). For gesture recognition, when the time delay was less than 0.08 s, the model could be employed in a real-time system [35]. Therefore, we think the dual-output model could be used in real-time. For the other two dual-output model benchmarks, their time delays were also less than 0.08 s (Table 1), so they could also be used in real-time systems.

### 4.8. Improvement of Dual-Output Model

A loss weighting method was used to define the total loss in our dual-output model, but this method introduced a hyperparameter, and the fixed weight ratio might not perform well due to the different decay rates of loss functions. Therefore, an adaptive weight variable could be introduced to replace the weight ratio.

Our model used an optimizer after defining the total loss. In practice, it might appear that no matter how the weight ratio was adjusted, only part of the losses was dominant, while the losses of other tasks did not work. In addition to dynamically adjusting the weight parameters, different learning rates could be assigned to the optimizers for different losses.

### 4.9. Benefits of the Dual-Output Model Compared with Stacked Single-Output Models

Compared with stacked two single-output models to obtain classification attribute and regression attribute, respectively, a dual-output model can receive two characteristics simultaneously by only one model. Benefits of the dual-output model compared with stacked single-output models are as follows:

1. Avoiding the analysis for the internal relationships of multi-model structure 

When stacking multiple single-output models, the output and input of different models and the connections between the models need to be considered. In addition, when numerous models are cascaded, the output of the previous model may affect the decision of the latter model, and the output of the latter model may also affect the results of the previous model, so the internal process and the final output can be very complicated. The multi-output model has only one model and does not need to consider the intermediate output, which saves a lot of intermediate analysis.

2. Tuning parameter and training model more conveniently

Some shallow learning models have multiple hyperparameters. The total hyperparameters will become more with stacked models. When stacked models are retrained, all hyperparameters need to retune, and the process can be very complicated. In a dual-output model, only one model requires tuning and training. Thus, the process is more convenient.

3. Performing tasks in a more effective way.

When used in practical tasks, the parallel output model is more effective in control than the sequential output model. For example, in prosthesis control, if a sequential model composed of multiple single-output models is used first to classify which finger to manipulate, and then predict the angle of the finger joint, the task decision-making process will be time consuming [36]. With the parallel output model simultaneously, the classification and regression tasks are solved at the same time, and the task decision-making process becomes more efficient.

### 4.10. Limitations and Future Work

Our model only achieved better results than the four model benchmarks and thus provided one promising solution in similar hand gesture recognition scenarios. However, future studies should be performed to see whether the proposed model can still perform best in other applications and data set. We only studied single finger movements and did not consider multiple coupled finger movements due to complex hand anatomy, which is common in practical tasks. Therefore, this is a potential limitation in our research. Future work would involve the coupling of multiple fingers. In addition, we only used modified barometric sensors to measure the muscle force of the wrist, so the information obtained was limited. EMG has been widely applied to hand gesture recognition. The fusion of multiple sensors could obtain more information and higher recognition accuracy. Future work would involve applying multiple sensors to a dual-output model. When re-wearing the wearable device, the positions of sensors may shift, resulting in a different distribution of the collected data. Future studies would involve the reproducibility of the device location.

### 4.11. Potential Applications

Although this paper only considered establishing a dual-output model, the research could potentially be extended to multiple outputs. For gesture recognition, in addition to hand gesture classification and finger angle estimation, the wrist angle could also be output simultaneously. In practical application, this method could be used in a controlled environment with both discrete and continuous variables: hand gesture classification for controlling discrete variables, and finger angle estimation for controlling continuous variables. For example, gesture recognition is used to control the direction and displacement of mobile robots. In this application, the direction is a discrete variable, and the displacement is a continuous variable. This method could pave the way for human–machine interaction [37], home automation [38], rehabilitation systems [39], robot control [40], consumer electronics [41], and other related applications.

## 5. Conclusions

In this paper, we presented a novel framework to enable both hand gesture classification and finger angle estimation simultaneously via a deep learning model with dual outputs. A dual-output deep learning model and a data augmentation method were proposed and evaluated by finger flexion/extension trials. The results show that the dual-output deep learning model can output hand gesture and finger angle at the same time and that data augmentation methods can improve the accuracy of results. This approach could be widely utilized in medical rehabilitation, virtual and augmented reality, sign language recognition, and other HCI-related applications.

## Figures and Tables

**Figure 1 sensors-20-02972-f001:**
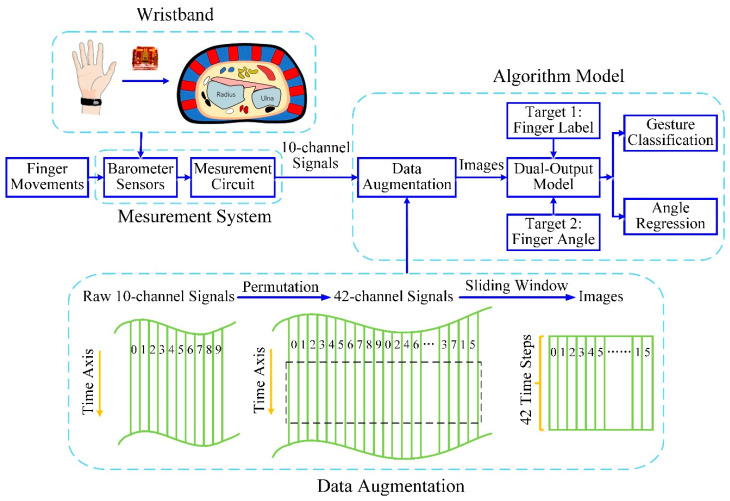
System framework of the novel dual-output convolutional neural network (CNN) model for simultaneous hand gesture recognition and finger angle estimation.

**Figure 2 sensors-20-02972-f002:**
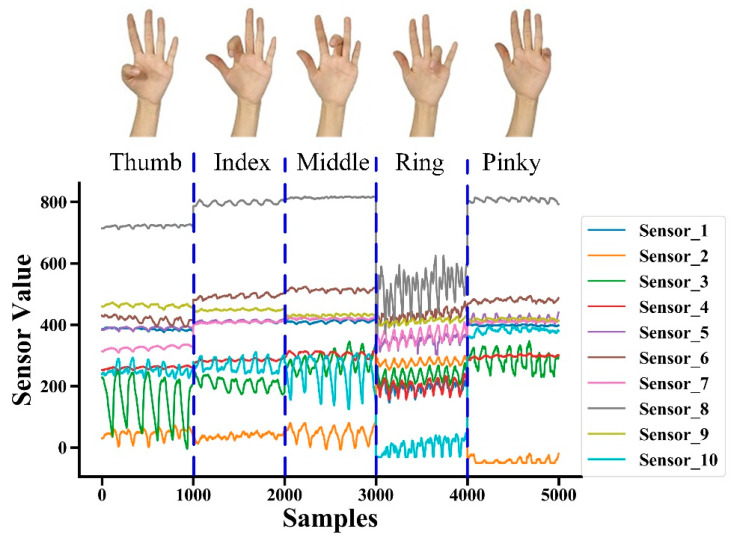
Raw signals from 10 modified barometric sensors for a typical trial (subject 1, trial 1) showing distinct patterns between each gesture.

**Figure 3 sensors-20-02972-f003:**
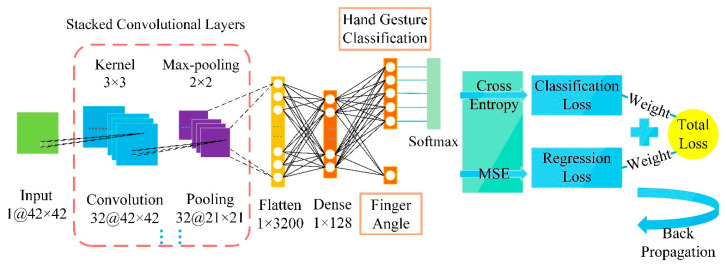
Dual-output CNN model structure diagram.

**Figure 4 sensors-20-02972-f004:**
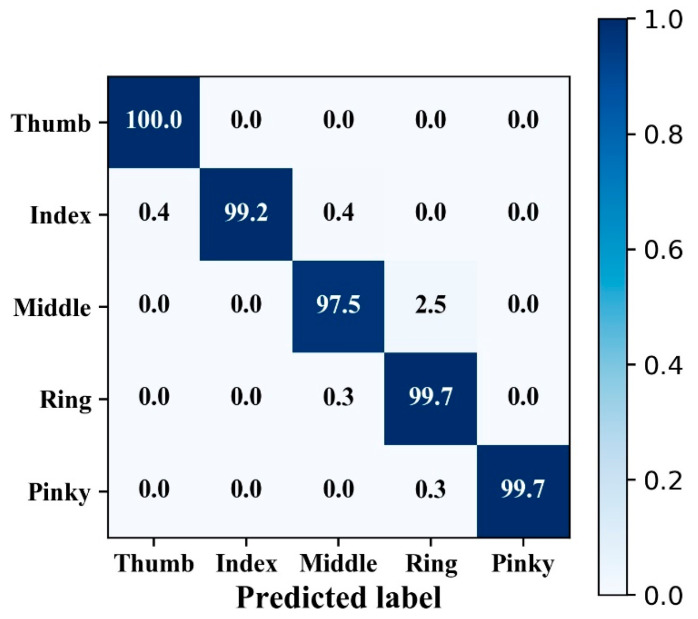
Confusion matrix of hand gesture classification from a typical subject (subject 10). The overall classification accuracy for this trial was 99.2%.

**Figure 5 sensors-20-02972-f005:**
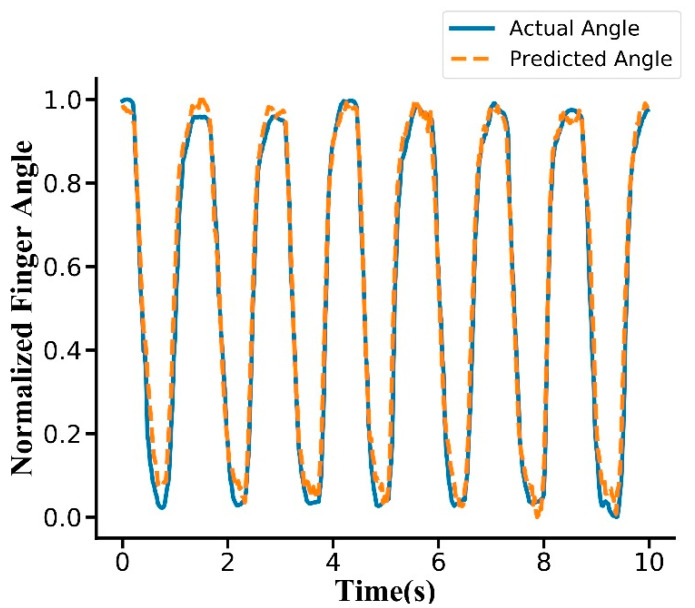
Typical trial showing actual and predicted normalized finger angles (subject 10, thumb). The $R^{2}$ score for this trial was 0.965. Finger angles were normalized from 0 and 1.

**Figure 6 sensors-20-02972-f006:**
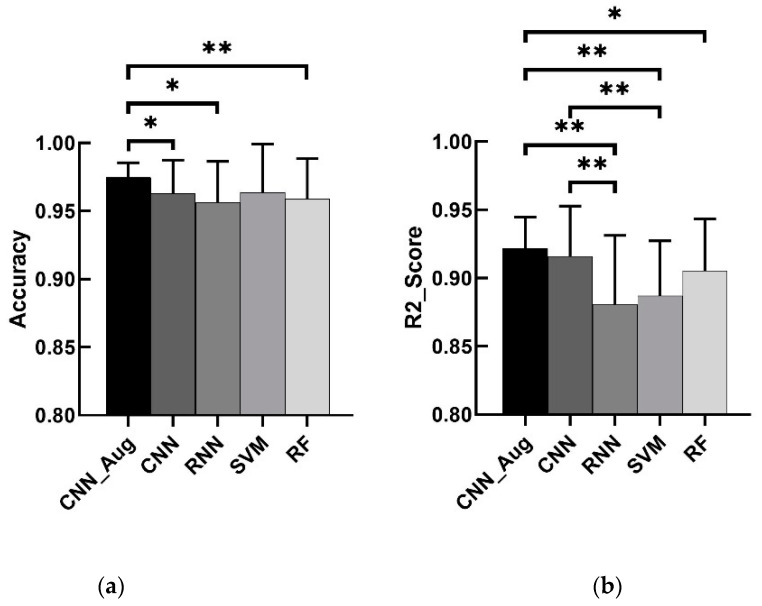
Overall results for different models: CNN_Aug (presented model) = convolutional neural network with data augmentation, CNN = convolutional neural network without data augmentation, RNN = recurrent neural network, SVM = support vector machine, and RF = random forest. (**a**) The results of hand gesture classification accuracy for different models. (**b**) The results of finger angle estimation fitting for different models. ** denotes p<0.001 and * denotes p<0.05.

**Figure 7 sensors-20-02972-f007:**
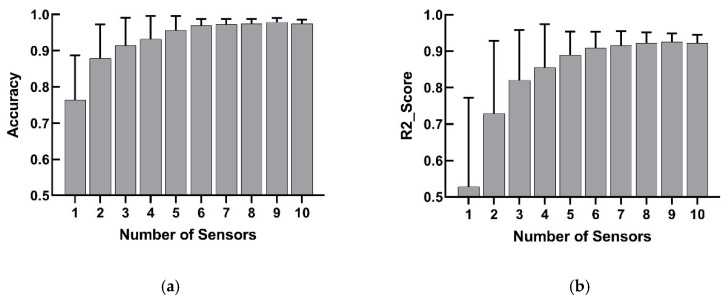
Overall results for different sensor numbers between 1 and 10. (**a**) The results of hand gesture classification accuracy for different sensor numbers. (**b**) The results of finger angle estimation fitting for different sensor numbers.

**Table 1 sensors-20-02972-t001:** Calculation time of three dual-output models.

Model	Time (s)
CNN with data augmentation	0.0444
CNN without data augmentation	0.0489
RNN	0.0424

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
