# Peer review of "Simultaneous Hand Gesture Classification and Finger Angle Estimation via a Novel Dual-Output Deep Learning Model"

_sensors, 2020, doi:10.3390/s20102972_

Round 1

Reviewer 1 Report

In general, the paper is very interesting and the authors contributed significantly to develop the model of deep learning
for the hand gesture classification and finger angle at the same time.
There are a few limitations in the study that authors should address before the final submission.

Major Comments:

The overall results of hand gesture classification and finger angle estimation fitting were 96.3% and 95.6% respectively.
There is minor accuracy difference b/w both models. Authors did not mention which model would be ultimately the best
for hand gesture classification and finger angle estimation and why?

Moreover, the LSTM model is best for the time-series signal whereas CNN is better for the data collected
for the images from optical sensors. Why authors preferred to implement CNN Dual-Output Model as compared to the LSTM model?

when LSTM has the ability to directly deal with the time-series signal. Moreover, it is also hard to explain the physical meaning of the features extracted from the signal converted into the images. which raised another challenge in healthcare applications.
For the Comparative Study b/w the different deep-learning models it is fine to compare the results. But claiming the CNN with the ultimate solution is not true. The authors should address these limitations.
it would be also appreciated if the author explains extracted features from the time-series signal and the signal converted in the images for CNN and a separate section to make it more clear.

English Proofreading is also mandatory before final submission.

Reviewer 2 Report

This paper proposes a CNN model with dual output to classify hand gestures and estimate finger joint angle by using the multi-channel FMG signal. Compared to the conventional learning approaches, the proposed data augmentation approach has advantages in the accurate classification and precise estimation. This paper is well organized. However, the reviewer has questions as follows.

- Originality
A similar research is published by the same research group [21]. Please clarify the originality of this paper compared to the previous paper.

There are several methods to simultaneously output the results of classification and estimation, for example [A][B]. Please refer to the conventional methods and clarify whats are the originality and advantage of the proposed method.
[A] Z. Abraham and P.N. Tan, An Integrated Framework for Simultaneous Classification and Regression of Time-Series Data, In Proc. the 2010 SIAM International Conference on Data Mining. Society for Industrial and Applied Mathematics, 2010.
[B] Zairan Wang et al., HCR-Net: A Hybrid of Classification and Regression Network for Object Pose Estimation, In Proc. the Twenty-Seventh International Joint Conference on Artificial Intelligence, 2018.

- Validity
Although the authors did a great effort on the comparison of the models, the performance metrics is not deeply discussed. Since our hand has more than 20-DoF, the 5 gestures used in the experiment are too simple to evaluate the classification performance. Please explain what kinds of gesture are important and show the performance. For example, fundamental classes of the hand posture is shown in [C].
[C] R.H. Liang and M. Ouhyoung. A sign language recognition system using hidden markov model and context sensitive search. In Proc. the ACM symposium on virtual reality software and technology. 1996.

Advantage of the proposed method is not clear. Although the proposed method showed high classification accuracy,
more accurate methods [25][26] are available as explained in section 4.1. Please show the benefit of the dual output.

Generally, multiple fingers are interlocked. Therefore, it is important to estimate each finger joint angle. However, only one finger joint angle is the output of the proposed system. Please use 5 finger joint angles as the estimation outputs.

- Reproducibility
It is not clear why the author used barometric sensors. The device proposed in the previous literature [20] has both EMG and FMG sensing functions. However, the authors used only FMG signal. Please explain why the EMG signals were not considered. It is also interesting that if the authors apply the proposed method to the EMG signal. Please give a comment.

The range of finger angle is not shown in this paper. Please explain the minimum and maximum angles.

- MISC
It is important to reveal why the data augmentation showed better classification accuracy compared to the conventional approaches. Related to this, the authors stated that this is because the data augmentation method took the hidden relationship between different sponsors and useful information contained in history data into account (in p.11 l.389). This explanation is not clear. Please show experimental data to explain the effectiveness.

Wearable device has a problem with the reproducibility of the device location. Please give a comment.

It is not clear that the proposed method can be used for the realtime application. Please show a calculation time.

Please give a discussion about the number of sensor elements. Is the proposed method scalable?

Reviewer 3 Report

An interesting approach to hand gesture classification and finger angle estimation.

  • How did you measure the angle between your fingers? What kind of data glove did you have? What is the accuracy of the measurements?
  • You can enhance all the figures Symbols of variables must be written with italic fonts.
  • What does the star mean in the figure 6? Explain the meaning of
  • It is useful if you can highlight any problems that you had in implementing the proposed method.

Round 2

Reviewer 1 Report

Accept in the present form with a bit improvement in the formatting of the tables and English proofreading. 

Author Response

Thank you for your comments. We have improved the table formatting and finished the English proofreading. 

Reviewer 2 Report

The authors have made careful revision by following the Reviewers’ comments. However, the following major concerns were not addressed by the authors' response.

1. Contributions
The authors stated that the previous method [1] was more complicated and less accurate compared with our proposed dual-output deep learning model.
However, the number of gesture types used in the previous study [1] is not same as this paper. Therefore, the comparison is not fair.

The authors argued that the advantages of the proposed method are 1) simple model structure and 2) simple model training and parameter adjustment.
However, the needs for the simplification is not explained.

2. Requirements
The dynamic recognition process of multiple fingers is very difficult because the flexion angles of different fingers are different, and it involves a lot of research on hand anatomy. Therefore, we chose 5 single fingers as the gesture set.
I agree with the difficulty of the dynamic sensing but the reason why the 5 types of motion are used is not provided.

It is also important to estimate multiple finger joint angles simultaneously. However, the requirement for the single angle sensing is not thoroughly discussed.

3. MISC
The author claimed that the number of sensors are investigated in the previous study [21]. However, the authors proposed a new model which is not investigated in the previous study.

The authors explained that the prediction time is about 1.48598 ÷ 100 × 2 = 0.003s. Because the prediction has overhead process for each gesture, the estimation is not convince.
